# Hazard Identification Related to the Presence of *Vibrio* spp., Biogenic Amines, and Indole-Producing Bacteria in a Non-Filter Feeding Marine Gastropod (*Tritia* *mutabilis*) Commercialized on the Italian Market

**DOI:** 10.3390/foods10112574

**Published:** 2021-10-25

**Authors:** Patrizia Serratore, Giorgia Bignami, Fabio Ostanello, Luna Lorito

**Affiliations:** Department of Veterinary Medical Science, University of Bologna, Viale A. Vespucci 2, 47042 Cesenatico, FC, Italy; patrizia.serratore@unibo.it (P.S.); giorgia.bignami@unibo.it (G.B.); luna.lorito2@unibo.it (L.L.)

**Keywords:** *T. mutabilis*, hazard identification, *Vibrio* spp., biogenic amines, indole-producing bacteria

## Abstract

*Tritia mutabilis* is a carrion-feeder edible marine gastropod with an open circulatory system. Therefore, biological, and chemical contaminants associated with the feed can reach all body tissues. The aim of the present study was to investigate the possible association of these characteristics with some food safety hazards. *Vibrio* spp. load, and the prevalence of pathogenic *V. parahaemolyticus*, *V. vulnificus*, and *V. cholerae*, were investigated. Moreover, biogenic amines (BAs) and indole-producing bacteria (IPB), markers of seafood decomposition, were quantified for the first time in an edible carrion-feeder. Overall, 49 batches were analyzed (38 from retail, and 11 from primary production). The *Vibrio* spp. load resulted of 5.64 ± 0.69 log_10_ CFU g^−1^ at retail, and 5.27 ± 0.74 at harvest but all batches resulted negative for pathogenic *Vibrio*. Histamine, putrescine, cadaverine, and tyramine were detected both at harvest and at the retail level. Their sum (BAs Index) showed a mean value of 50.45 and 65.83 mg Kg^−1^ in batches at harvest and at retail, respectively. IPB were detected at harvest and upon refrigeration for three days (T1–T3). The mean load resulted in 2.52 ± 0.85 log_10_ MPN g^−1^ at T0, 3.31 ± 1.23 at T3 in batches immediately refrigerated, and 3.22 ± 1.18 at T3 in batches previously immersed in clean seawater. Our results contribute to identifying food-borne hazards for *T. mutabilis* that may be related to the retention of biogenic amines and indole-producing bacteria due to carrion feeding.

## 1. Introduction

Fishing of marine gastropod *Tritia mutabilis* is by far the most important activity carried out by artisanal fisheries in the Central and Northern Adriatic Sea, yielding from 2000 to 3000 tons of landings each year [1,2].

According to diet, gastropods can be herbivorous, carnivorous, and omnivorous, and their feeding strategies are also very diversified including grazing, suspension-feeding, predation, and even parasitism [3]. *T. mutabilis* is a scavenger gastropod, feeding on carrion in the adult stage [2,4].

Gastropods, together with bivalve mollusks, possess an open circulatory system, characterized by irregular spaces or sinuses within the tissues between the distributing and collecting blood vessels [5]. The system is considered open because the blood (hemolymph) empties from a contractile heart and major supply vessels into the body cavity (hemocoel), where it directly bathes the organs [6]. Therefore, contaminants including bacteria, viruses, and chemicals circulating after feeding can easily reach all body tissues.

The genus *Vibrio* comprises the major culturable bacteria in marine and estuarine environments [7], and with 130 confirmed species, it represents one of the most diverse marine bacterial genera, geographically spread all over the world, but more commonly occurring in warmer waters [8]. The genus *Vibrio* includes several food-borne pathogens that cause a spectrum of clinical conditions, that can be grouped into three major syndromes: gastroenteritis, wound infection and septicemia [9]. Pathogenic vibrios have emerged as new threats since the 1970s [10], causing an increased incidence of human infections worldwide [11,12,13], and the consumption of raw or undercooked bivalve mollusks is considered the primary source particularly oyster, clam, and mussels, because of their filter-feeding habit that allows concentrating particles associated with various contaminants from the surrounding waters [14,15,16]. Unfortunately, marine gastropods, unlike bivalves, are poorly investigated in terms of food safety.

Besides the so-called “big four”, namely *V. cholerae, V. parahaemolyticus*, *V. vulnificus*, and *V. alginolyticus*, other *Vibrio* species have been associated with food-borne illness, including *V. fluvialis*, *V. mimicus*, *Grimontia* (*Vibrio*) *hollisae*, and *V. metschnikovii* [8].

Nevertheless, the European Union (EU) legislation does not include pathogenic vibrios, biogenic amines and other products of the decomposition among the safety criteria for gastropods, regarded as equivalent to bivalve mollusks, echinoderms, and tunicates [17]. To our best knowledge, no outbreaks of food poisoning caused by *T. mutabilis* have ever been reported.

Appropriate cooking treatment is effective to inactivate microorganisms associated with seafood, but it is considered little or nothing effective on their toxic products, as documented for many marine biotoxins [18,19], biogenic amines (BAs) [20], and indole, whose production is fostered by high pH and temperature [21].

BAs are organic bases with aliphatic, aromatic, and heterocyclic structures produced mainly by bacteria [20]. During the decomposition of seafood, various amounts of BAs are usually produced, in particular, the Gram-negative bacteria are considered the predominant amine-forming bacteria in fish, cephalopods, and shellfish [22], including *Vibrio* spp. [23]. BAs are formed mainly by decarboxylation of specific free amino acids or by amination and transamination of ketones and aldehydes [23,24]. Decarboxylation of histidine, lysine, ornithine, and tyrosine results in the production of histamine, cadaverine, putrescine, and tyramine respectively [25], which are commonly used as chemical indicators of fish decomposition [22]. BAs in food constitutes a potential public health concern due to their physiological and toxicological effects, such as histamine intoxication, migraine, and food intolerance crises [26]. Except for histamine in fishery products no threshold is established for these biomolecules in EU legislation. On the other hand, despite a widely reported association between histamine and scombroid food poisoning, histamine alone appears to be insufficient to cause food toxicity, as other BAs, i.e., putrescine and cadaverine, may potentiate histamine toxicity [27].

Indole is widespread in nature because various species of bacteria can produce a considerable amount of this heterocyclic aromatic amine [28], via the action of the enzyme tryptophanase, expressed in many Gram-negative, as well as Gram-positive bacteria [29]. It should be noted that members of the genus *Vibrio* are nearly all indole positive [30]. Indole may serve as a chemical indicator for the evaluation of incipient spoilage of crustaceans and other seafood products [31] and has long been regarded as a typical N-heterocyclic aromatic pollutant due to its toxicity and potential mutagenicity causing animal hemolysis, hemoglobinuria nephrosis, temporary skin irritation and tumor formation [21]. Normally, indole is metabolized to indoxyl sulfate in the liver and then excreted into the urine, but if the dietary intake exceeds the renal clearance, the serum levels of indoxyl sulfate increase with toxicological effects [2].

Official controls on non-filter feeding marine gastropods harvested from non-classified production areas are considered not yet fully implemented by the EU Member States [32], in any case, several food-borne hazards are not considered by the Legislation in force. Among them, pathogenic vibrios, BAS and indole.

The aim of the present study was to ascertain possible food-borne hazards related to the accumulation of bacteria and chemical products of the decomposition in *T. mutabilis* from Emilia-Romagna Region, Northern Italy. Batches at harvest and retail were considered under different conditions for viability of the batches, *Vibrio* spp. load, and the prevalence of pathogenic species, namely *V. cholerae*, *V. parahaemolyticus*, and *V. vulnificus*. To ascertain the possible accumulation of toxic catabolites assumed from the carrion, Biogenic Amines (BAs) and Biogenic Amines Index (BAI) were investigated at harvest and retail, whereas Indole Producing Bacteria (IPB) load was investigated at harvest and under refrigerated storage.

## 2. Materials and Methods

The study area is part of the FAO zone 37.2.1, GSA (Geographical Subarea) 17, along the Emilia-Romagna coastline, Adriatic Sea, Northern Italy (Figure 1). 

*T. mutabilis* is not available year-round, because the fishing season is from the early autumn to the late spring. Therefore, the study required several surveys carried out between October 2017 and February 2021.

The research program was divided into three Work Packages (WPs): (WP1) evaluation of batches viability, *Vibrio* spp. load and the prevalence of pathogenic vibrios at retail; (WP2) evaluation of batches viability, *Vibrio* spp. load, and the prevalence of pathogenic vibrios and BAs content at retail and at harvest; (WP3) evaluation of batches viability, *Vibrio* spp. load, prevalence of pathogenic vibrios and IPB load at harvest and during refrigeration at 4 °C with and without prior treatment of immersion in clean seawater in a Recirculating Aquaculture System (RAS). The treatment is normally intended for bivalve mollusks depuration (to reduce *E. coli* contamination) and conditioning (to remove sand, mud or slime, to improve organoleptic qualities and to ensure good vitality).

The batches from primary production were obtained directly from a fisherman (at landing), and those at retail from different fish shops. All batches were transported immediately to the laboratory in isothermal boxes, stored at 6–8 °C to avoid *Vibrio* spp. entering a viable but non-culturable state, and processed within 24 h. 

### 2.1. Viability of the Batches (WPs 1, 2, and 3)

According to the EU legislation [17], “fresh” marine gastropods must be sold alive as bivalve mollusks, to which they are assimilated. They must show characteristics of freshness and viability, must be responsive to the touch by closing their open valves, and must contain normal amounts of intravalvular liquid. This approach is tailored to the specific features of bivalves, and obviously, it is unsuitable for gastropods, because they lack two valves and obviously, intravalvular liquid. The viability of the batches of *T. mutabilis* was ascertained following an internal protocol, by spreading table salt on the animals, as NaCl results in an aversive stimulus promoting extroflexion of the foot and mucus production in viable animals [33]. For each batch, the viability was expressed as the percentage of living animals on the whole batch.

### 2.2. Abundance of Vibrio spp. and Pathogenic Vibrios (WPs 1, 2, and 3)

The abundance of Vibrio spp. was checked on thiosulfate-citrate-bile salts-sucrose (TCBS) agar (Oxoid, Milan, Italy) NaCl 3% (final) by the spread plate method at 20 °C for 3–5 days, following an internal protocol [34]. 

The sample units (10–20 individuals) were prepared according to a slight modification of the ISO 6887-3:2017/Amd.1:2020 method. Briefly, gastropods were rinsed with sterilized seawater, then their shells were cut aseptically to obtain 10 g of the whole body. Each unit was added to 90 mL of saline solution NaCl 3% and homogenized utilizing a rotary blender at medium speed. From this first dilution, further ten-fold serial dilutions were prepared with saline solution NaCl 3%.

From each homogenate and its ten-fold serial dilutions, 100 µL were spread plated on the agar plates and incubated at 20 °C. The results were expressed as log_10_ Colony Forming Units (CFU) g^−1^.

*V. parahaemolyticus*, *V. cholerae* and *V. vulnificus* were investigated utilizing the homogenate of each sample unit, from which 100 µL were spread plated on agar plate of CHROMagar™ Vibrio (CAV) (PBI International, Milan, Italy), then incubated at 37 °C for 24 h. Suspected colonies (*V. parahaemolyticus* mauve, *V. vulnificus*/*V. cholerae* green-blue to turquoise blue, *V. alginolyticus* colorless) were submitted to biochemical screening and genotyped by Polymerase Chain Reaction (PCR). Briefly, after a biochemical screening to ascertain the characteristics of the genus (pleomorphic Gram-negative rods, oxidase-positive, able to reduce nitrate, glucose-fermenting, sensitive to the vibriostatic O129/150 μg), suspected strains were genotyped by PCR as previously described [35] to test their respective species-specific and pathogenic gene markers. For *V. parahaemolyticus tox*R, *tdh* and *trh*; for *V. cholerae tox*R, *ctx* and *stn*/*sto*; for *V. vulnificus* only the species-specific gene marker *vvh*A, because, according to the Food and Agricultural Organization of the United Nations and the World Health Organization [36], all *V. vulnificus* strains may be prudently considered virulent. The oligonucleotide primers and the PCR conditions are reported in Appendix A.

### 2.3. Biogenic Amines (WP2)

BAs were quantified by an HPLC method employing a UV detector after derivatization by Dansyl chloride. Analyses were performed by an external service (FoodMicroTeam s.r.l., Academic Spin-off of the University of Florence, Italy), and samples were supplied frozen (−20 °C). Considering the results for each batch, the sum of histamine, putrescine, cadaverine, and tyramine quantities were calculated and expressed as BAI (Biogenic Amines Index) according to Veciana-Nogués et al. [37].

### 2.4. Abundance of IPB (WP3)

IPB load was determined in *T. mutabilis* by Most Probable Number procedure, indicated by Leitão and Rios [38] as the most reliable approach for the quantitative evaluation of the indole positive microflora. Three series of three tubes containing 5 mL of tryptone broth 1% were prepared. One series was inoculated with 0.5 mL of the homogenate (dilution 1:10), and the remaining two with 0.5 mL of two further ten-fold serial dilutions each. After incubation at 20 °C for 3–5 days, Kovac’s reagent was added to each tube. The reagent builds with indole a cherry-red complex, allowing to confirm the enzymatic activity of tryptophanases and the positivity for IPB. 

### 2.5. Statistical Analysis 

Statistical analysis was performed on *Vibrio* spp. load, BAs, BAI, and IPB content of the batches of *T. mutabilis*. According to the EU legislation [17], bivalve mollusks and gastropods placed on the market must be alive, but a mortality ≤10% is almost always accepted. For this reason, batches were divided into two groups based on the percentage of viability: high viability (≥90%) and low viability (<90%). The one-sample Kolmogorov–Smirnov (K–S) test was performed to assess data distribution. According to K–S test results, either non-parametric Mann–Whitney U test (M–W U) or Student’s *t*-test were used to establish possible differences in *Vibrio* spp. load, BAs, BAI, and IPB content between the two viability groups or between treatment groups (with or without immersion). Paired samples *t*-test was used to compare viability, *Vibrio* spp. load and IPB content between T0 and T3 within treatment groups.

Statistical significance was set at *p* ≤ 0.05. All statistical analyses were performed using the software SPSS 26.0.0 (IBM SPSS Statistics, Armonk, NY, USA).

## 3. Results

Overall, forty-nine batches of *T. mutabilis* were processed. Results are reported below for each of the three WPs. All batches were negative for *V. parahaemolyticus*, *V. vulnificus*, and *V. cholerae*.

### 3.1. Viability of the Batches and Vibrio spp. Load at Retail (WP1)

The twenty-four batches of *T. mutabilis* from retail showed different values of viability (Table 1). Only 20.8% (5/24) of the batches showed 90–100% of viability, whereas 16.7% (4/24) of the batches had 70–80% of viability, 41.7% (10/24) had 19–50%, and 20.8% (5/24) had 0%. *Vibrio* spp. load resulted in a variable within 4.07 and 7.69 log_10_ CFU g^−1^ with a mean value of 5.70 ± 0.75 log_10_ CFU g^−1^. No statistically significant difference (*p* > 0.05) of *Vibrio* spp. load was observed between the group with viability ≥ 90% (5.27 log_10_ CFU g^−1^) and the group with viability < 90% (5.81 log_10_ CFU g^−1^).

### 3.2. Viability of the Batches, Vibrio spp. Load and BAs Content at RETAIL and at Harvest (WP2)

Overall, 18 batches of *T. mutabilis* were analyzed (Table 2). The four batches from primary production showed viability of 100%. Two of the fourteen batches from retail had a viability of 90%, ten < 90% and two 0%. *Vibrio* spp. mean load resulted in 5.60 ± 0.61 log_10_ CFU g^−1^ in the batches from retail, and 5.30 ± 0.85 log_10_ CFU g^−1^ in the batches at harvest. Histamine, putrescine, cadaverine, and tyramine were detected in all samples and each BAI (their sum) was calculated. All BAI value resulted >15 mg Kg^−1^, and 50% (9 out of 18) resulted >50 mg Kg^−1^ (min-max: 23.2–225.1 mg Kg^−1^). The mean value resulted in 50.45 ± 21.27 Kg^−1^ in the four batches from the primary production, and 65.83 ± 48.36 mg Kg^−1^ in the fourteen batches from retail.

No statistically significant difference of *Vibrio* spp. load, histamine, putrescine, cadaverine, tyramine, and BAI quantity were observed between batches from the primary production or from retail.

### 3.3. Viability of the Batches, Vibrio spp. and IPB Load at Harvest and during Refrigeration (WP3)

Seven batches at harvest were utilized to verify any influence of the refrigerated storage at 4 °C for three days (T1, T2, T3) on the viability of the batches, *Vibrio* spp. load and IPB load. Three batches were immediately refrigerated, while four batches were partly immediately refrigerated and partly refrigerated after 18–24 of immersion in clean seawater in a RAS. Results are reported in Table 3.

Overall, at harvest (T0) the mean value of viability resulted in 98.63 ± 3.63% (six batches 100% and one batch 90.4%). The mean value of viability at T3 resulted in 62.43 ± 26.95% in the samples immediately refrigerated and 94.25 ± 7.59% in the samples previously immersed in clean seawater. The mean load of *Vibrio* spp. resulted in 5.08 ± 0.82 log_10_ CFU g^−1^ at T0, 5.10 ± 0.62 at T3 without immersion, and 5.46 ± 0.12 at T3 with previous immersion. 

IPB load resulted in 2.52 ± 0.85 log_10_ MPN g^−1^ at T0, 3.31 ± 1.23 and 3.22 ± 1.18 at T3 without and with previous immersion, respectively. 

In batches refrigerated after 18–24 of immersion in clean seawater in a RAS, no significant differences (*p* > 0.05) were observed in viability, *Vibrio* spp. load and IPB content between T0 and T3. Similarly, in batches refrigerated without immersion, no significant differences (*p* > 0.05) were observed in *Vibrio* spp. load and IPB content between T0 and T3. However, viability at T3 (62.4%) was significantly lower (paired-*t*: 3.94; *p* = 0.008) than that observed at T0 (98.6%). 

## 4. Discussion

The EU legislation assimilates the marine gastropods to the bivalve mollusks in terms of safety criteria for human consumption but allows the harvesting of gastropods that are not filter-feeders in non-classified areas [39,40]. This provision relies on earlier statements expressed in Regulation UE 558/2010 [41]: the risk of accumulation of microorganisms related to fecal contamination may be considered remote in animals that are not filter-feeders; no epidemiological information was reported to link the provisions for classification of production areas with risks for public health associated with marine gastropods which are not filter feeders. 

Bivalve mollusks and gastropods have both an open circulatory system [5], and therefore bacteria and other contaminants can be expected in all organs and tissues, independently of the different feeding habits. Despite this biological evidence, the microbiology of marine gastropods has been poorly investigated, except few studies concerning disease outbreaks of Abalone, reporting high bacterial contamination of tissues also in healthy animals [42], and the study of Cheng et al. [43] reporting high contamination of the gastropod *Niotha clathrata* muscle by *Vibrio* spp. (6–8 log_10_ CFU g^−1^).

A recent study evidenced a *Vibrio* spp. mean load exceeding 5 log_10_ CFU g^−1^ in the tissues of *T. mutabilis* and 5.79 log10 CFU g^−1^ in *Bolinus brandaris*, another carnivorous gastropod fished, commonly by-catch, in the Adriatic Sea [33]. 

Accordingly, in the present study *Vibrio* spp. mean load resulted in >5 log_10_ CFU g^−1^ in all the three WPs. In WP1 resulted in 5.70 ± 0.75 log_10_ CFU g^−1^ in batches from retail (Table 1). In WP2 resulted in 5.60 ± 0.61 log_10_ CFU g^−1^ in the batches from retail, and 5.30 ± 0.85 log_10_ CFU g^−1^ in the batches from primary production (Table 2). In WP3 resulted in 5.08 ± 0.82 log_10_ CFU g^−1^ in the batches from primary production (T0 = before refrigeration), 5.10 ± 0.62 log_10_ CFU g^−1^ after refrigeration (T3) without previous treatment of immersion, and 5.46 ± 0.12 log_10_ CFU g^−1^ after refrigeration (T3) with previous treatment of immersion (Table 3). It should be noted that these values result higher than those ascertained in *R. philippinarum* in the same area, showing a mean load of 4.69 ± 0.65 log_10_ CFU g^−1^ in a multi-year retrospective study [44], and 4.82 ± 0.71 log_10_ CFU g^−1^ in a concurrent project (unpublished data). 

Notwithstanding the high *Vibrio* spp. load, all batches of *T. mutabilis* resulted negative for pathogenic *V. parahaemolyticus*, *V. vulnificus* and *V. cholerae*, whereas batches of *R. philippinarum* showed 62% of positivity for at least one of the targets in a concurrent project (unpublished data), as proof of their circulation in the area. The discrepancy between the two species is difficult to explain and needs further investigation. It can only be assumed that the microbial environment and the chemical characteristics of the carrion on which *T. mutabilis* feed could result in an unsuitable environment for *V. parahaemolyticus*, *V. vulnificus* and *V. cholerae*. 

The fishing of *T. mutabilis* is practiced within three miles of the coastline by dedicated traps with dead fishes as baits, predominantly bluefish. These gastropods use their sense of smell to find carrion, therefore spoiled fish is preferred by some fishermen. Other fishermen prefer thawed white fish of acceptable quality to prevent the bitterness of gastropod flesh. 

Given the feeding habit of *T. mutabilis*, chemical products of the decomposition should be considered of concern, particularly BAs and indole. However, to our best knowledge, this is the first study conducted on a marine scavenger gastropod and comparative data are lacking. Histamine, putrescine, cadaverine, and tyramine, were always detected (Table 2), including 14 batches of *T. mutabilis* from retail and four batches from primary production. Considering the BAI limit of acceptability suggested for tuna (<50 mg Kg^−1^) and for anchovy, (<15–16 mg Kg^−1^) [25], the BAI values registered in *T. mutabilis* may be considered unquestionably high. All samples had a value >15 mg Kg^−1^, and 50% (9 out of 18) had a value >50 mg Kg^−1^ (minimum 23.2 mg Kg^−1^, maximum 225.1 mg Kg^−1^). While not differing significantly, the BAI mean value resulted higher in the batches from retail (65.83 mg Kg^−1^) than in the batches from the primary production (50.45 mg Kg^−1^) suggesting that, after the harvesting, endogenous production of BAs may occur over time, increasing the total amount of these substances in *T. mutabilis* tissues.

Indole may serve as a chemical indicator for the evaluation of incipient spoilage of crustaceans and other seafood products [31]. A limit of 250 μg Kg^−1^ was established by the US Food and Drug Administration, and in several countries, to identify shrimps in the first stage of decomposition, but it was demonstrated that also shrimps with a level of 44.6 μg Kg^−1^ show the rejection conditions [45]. Almost all species of *Vibrio* spp. can produce indole via tryptophanase activity [30,46], and the activity of tryptophanase may result increased 21-fold at pH 8–9 [47]. Marine carrion is rich medium in a strongly alkaline environment (pH > 8), and it may be expected to contain a significant amount of this substance. As shown in Table 3, IPB were always detected in *T. mutabilis* batches at harvest (mean load 2.52 ± 0.85 log_10_ MPN g^−1^), and with increasing value after three days of refrigeration (3.31 ± 1.23 and 3.22 ± 1.18 log_10_ MPN g^−1^ without and with immersion respectively) suggesting that IPB are predominantly represented by psychrophilic bacteria metabolically active at chilling temperature (4 °C). Considering that fish meat is naturally rich in tryptophan [48] this IPB load in a carrion-feeder is certainly potentially able to produce a high level of indole.

The immersion in clean seawater is provided by EU legislation for bivalve mollusks [17], but not for marine gastropods. Conversely, our results suggest that this treatment may ensure higher viability to *T. mutabilis* at retail, as the batches immersed in clean seawater after landing showed higher viability after three days of refrigeration (mean value 94.25%) than those immediately refrigerated (mean value 62.43%). It can be assumed that the low percentage of viability registered for 24 batches at retail (only 20.8% with 90–100% of viability) could be due to the lack of treatment of immersion after landing.

## 5. Conclusions

The feeding habit of *T. mutabilis* and its open circulatory system allows a high accumulation of bacteria (*Vibrio* spp.) and products of the decomposition associated with the carrion on which the animal feed, i.e., BAs and IPB. Notwithstanding the circulation of pathogenic vibrios in the area, namely *V. parahaemolyticus*, *V vulnificus*, and *V. cholerae* (high percentage of positivity in *R. philippinarum*) all samples resulted negative. To our knowledge, the present study is the first to address various issues of food safety involving *T. mutabilis*. For this species, the undiscovered safety issues seem to rely on the retention of biogenic amines and indole-producing bacteria because of the feeding on carrion. Obviously, the utilization of spoiled fish as bait should be discouraged, to prevent an excessive intake of products of the decomposition in *T. mutabilis*. Moreover, the lack of immersion in clean seawater after landing could be at the origin of the low viability of the batches at retail, with possible endogenous production of BAs.

## Figures and Tables

**Figure 1 foods-10-02574-f001:**
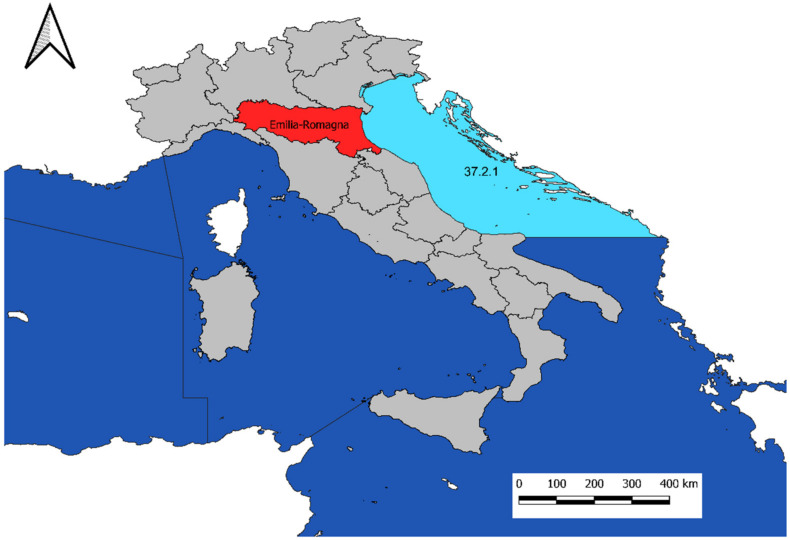
Sampling area along the Emilia-Romagna region coast, Northern Italy (red) Adriatic Sea (blue), FAO zone 37.2.1, GSA 17 (light blue).

**Table 1 foods-10-02574-t001:** Viability of the *Tritia mutabilis* batches at retail and *Vibrio* spp. load.

Viability Class	Batch id. (Date of Collection or Purchase)		Viability (%)	*Vibrio* spp. (log_10_ CFU g^−1^)
<90%	1379	(10 October 2017)	0	7.69
	1442	(26 February 2019)	0	6.16
	1446	(7 March 2019)	0	5.48
	1448	(18 March 2019)	0	6.26
	1453	(9 April 2019)	0	5.66
	1384	(25 October 2017)	19	6.42
	1407	(18 April 2018)	20	5.26
	1433	(15 January 2019)	20	4.34
	1447	(12 March 2019)	20	5.29
	1443	(28 February 2019)	25	5.18
	1452	(3 April 2019)	25	5.76
	1434	(17 January 2019)	30	5.92
	1438	(5 February 2019)	30	6.26
	1435	(22 January 2019)	40	6.29
	1440	(13 February 2019)	50	5.71
	1406	(21 March 2018)	70	6.32
	1439	(13 February 2019)	70	5.73
	1450	(21 March 2019)	70	5.79
	1449	(19 March 2019)	80	4.88
mean	-		29.95	5.81
std. deviation	-		26.64	0.71
std. error of mean	-		6.11	0.16
≥90%	1436	(29 January 2019)	90	5.03
	1437	(31 January 2019)	90	6.07
	1378	(4 October 2017)	100	6.00
	1382	(18 October 2017)	100	5.20
	1451	(25 March 2019)	100	4.07
mean	-		96.0	5.27
std. deviation	-		5.48	0.82
std. error of mean	-		2.45	0.37
all viability classes				
mean	-		43.71	5.70
std. deviation	-		36.21	0.75
std. error of mean	-		7.39	0.15

**Table 2 foods-10-02574-t002:** Viability of the batches of *Tritia mutabilis*, *Vibrio* spp. load, single BAs content, and BAI (Histamine + Putrescine + Cadaverine + Tyramine) at harvest and at retail.

Batch Origin	Batch id. (Date of Collection or Purchase)	Viability (%)	*Vibrio* spp. (log_10_ CFU g^−1^)	Histamine (mg Kg^−1^)	Putrescine (mg Kg^−1^)	Cadaverine (mg Kg^−1^)	Tyramine (mg Kg^−1^)	BAI (mg Kg^−1^)
Harvest	1389 (21 November 2017)	100	5.92	15.60	38.30	13.30	6.50	73.70
	1470 (12 February 2020)	100	5.53	19.00	5.10	17.10	5.50	46.70
	1471 (19 February 2020)	100	5.7	3.80	15.80	33.50	5.10	58.20
	1472 (24 February 2020)	100	4.04	3.90	6.30	8.70	4.30	23.20
	mean	100	5.30	10.58	16.38	18.15	5.35	50.45
	std. deviation	0.00	0.85	7.89	15.38	10.79	0.91	21.27
	std. error of mean	0.00	0.43	3.94	7.69	5.40	0.46	10.64
Retail	1406 (21 March 2018)	70	6.32	6.00	14.50	15.90	5.70	42.10
	1433 (15 January 2019)	20	4.34	2.00	9.60	19.10	3.30	34.00
	1434 (17 January 2019)	30	5.92	5.40	16.90	21.10	4.60	48.00
	1435 (22 January 2019)	40	6.29	7.00	2.96	43.60	6.70	60.26
	1436 (29 January 2019)	90	5.03	4.00	12.60	27.90	5.40	49.90
	1437 (31 January 2019)	90	6.07	2.30	16.00	27.90	4.60	50.80
	1438 (5 February 2019)	30	5.48	2.50	172.00	24.90	25.70	225.10
	1439 (13 February 2019)	70	5.73	4.60	19.70	19.30	2.50	46.10
	1442 (26 February 2019)	0	6.16	16.90	24.10	36.50	13.20	90.70
	1443 (28 February 2019)	25	5.18	1.70	8.00	12.60	8.50	30.80
	1446 (7 March 2019)	0	5.48	5.90	18.20	26.40	9.60	60.10
	1447 (12 March 2019)	20	5.29	5.00	8.20	16.30	7.80	37.30
	1448 (18 March 2019)	0	6.26	4.20	20.70	27.40	10.90	63.20
	1449 (19 March 2019)	80	4.88	39.00	8.10	28.50	7.70	83.30
	mean	40.36	5.60	7.61	25.11	24.81	8.30	65.83
	std. deviation	33.31	0.61	9.78	42.69	8.36	5.82	48.36
	std. error of mean	8.90	0.16	2.61	11.41	2.24	1.55	13.09
Total	mean	53.61	5.53	8.27	23.17	23.33	7.64	62.41
	std. deviation	38.72	0.65	9.26	38.07	9.07	5.25	44.23
	std. error of mean	9.13	0.15	2.18	8.97	2.14	1.24	10.43

**Table 3 foods-10-02574-t003:** Viability of the batches of *Tritia mutabilis*, *Vibrio* spp. load, and IPB load. Batches at harvest (T0) and refrigeration at 4 °C. T1–T3 = day of refrigeration with and without immersion.

Batch id.	Time	*Vibrio* spp. (log_10_ CFU g^−1^)		Viability of the Batches (%)		IPB (log_10_ MPN g^−1^)	
(Date of Collection or Purchase)		Without Immersion	With Immersion	Without Immersion	With Immersion	Without Immersion	With Immersion
1470 (12 February 2020)	0	5.53	-	100	-	1.66	-
	1	4.74	-	74.0	-	2.58	-
	2	5.71	-	32.0	-	5.04	-
	3	4.00	-	73.0	-	4.63	-
1471 (19 February 2020)	0	5.70	-	100	-	3.04	-
	1	5.49	-	86.0	-	4.04	-
	2	6.12	-	44.0	-	4.18	-
	3	5.08	-	54.0	-	5.18	-
1472 (24 February 2020)	0	4.04	-	100	-	2.18	-
	1	4.41	-	73.0	-	3.18	-
	2	4.67	-	69.0	-	3.63	-
	3	5.38	-	59.0	-	3.20	-
1486 (19 October 2020)	0	6.23	(6.03) ^a^	100	(100)	4.04	(3.30)
	1	4.63	5.79	100	100	2.88	2.92
	2	5.25	4.64	94.0	100	4.18	2.63
	3	5.88	5.53	84.0	93.0	2.86	3.38
14893 (November 2020)	0	4.98	(6.08)	90.4	(99.0)	2.81	(3.66)
	1	4.00	4.76	80.0	84.0	2.04	1.36
	2	5.64	5.36	35.0	84.0	4.18	2.63
	3	4.60	5.43	16.0	84.0	1.56	1.56
1492 (18 January 2021)	0	4.99	(4.92)	100	(100)	1.63	(1.63)
	1	5.09	5.23	100	100	1.36	2.97
	2	5.75	5.86	97.0	100	3.04	3.15
	3	5.45	5.58	51.0	100	2.54	3.63
1495 (22 February 2021)	0	4.09	(4.91)	100	(100)	2.30	(3.66)
	1	3.81	5.48	100	100	3.81	1.88
	2	4.86	5.87	100	100	2.88	2.81
	3	5.30	5.31	100	100	3.18	4.32
Total	mean at T0	5.08	(5.48)	98.63	(99.75)	2.52	(3.06)
	std. deviation at T0	0.82	0.66	3.63	0.50	0.85	0.97
	std. error of mean at T0	0.31	0.33	1.37	0.25	0.32	0.48
	mean at T3	5.10	5.46	62.43	94.25	3.31	3.22
	std. deviation at T3	0.62	0.12	26.95	7.59	1.23	1.18
	std. error of mean at T3	0.23	0.06	10.19	3.79	0.47	0.59

^a^ Values registered after immersion are reported in brackets.

## Data Availability

Not applicable.

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
