# Peer review of "Hazard Identification Related to the Presence of Vibrio spp., Biogenic Amines, and Indole-Producing Bacteria in a Non-Filter Feeding Marine Gastropod (Tritia mutabilis) Commercialized on the Italian Market"

_foods, 2021, doi:10.3390/foods10112574_

Round 1
Reviewer 1 Report
General comments
The work deals with the safety assessment of the commercialized non-filter feeding marine gastropod Tritia mutabilis as regards the presence of Vibrio spp,, biogenic amines and indole-producing bacteria. The detection of pathogenic species of the Vibrio spp. was also investigated. The manuscript is generally well written in good order, but an English editing is still needed. The methodology is clear for repetition and was logically designed. The statistical analysis, interpretation and discussion of the results were correct. It is a work with practical information on food safety control. Such results could be still useful for food establishments, scientific community and other interested third parties.
Author Response
General comments
The work deals with the safety assessment of the commercialized non-filter feeding marine gastropod Tritia mutabilis as regards the presence of Vibrio spp,, biogenic amines and indole-producing bacteria. The detection of pathogenic species of the Vibrio spp. was also investigated. The manuscript is generally well written in good order, but an English editing is still needed. The methodology is clear for repetition and was logically designed. The statistical analysis, interpretation and discussion of the results were correct. It is a work with practical information on food safety control. Such results could be still useful for food establishments, scientific community and other interested third parties.
Answer: The Authors thank the Reviewer for the positive comments on the paper. A native English speaker has proofread our English writing.
Reviewer 2 Report
In this manuscript entitled "THazard identification related to the presence of Vibrio spp., biogenic amines, and indole producing bacteria in a non-filter feeding marine gastropod (Tritia mutabilis) commercialized on the Italian market", the authors examined possible foodborne hazards related to the accumulation of bacteria and chemical products of the decomposition in T. mutabilis from Emilia-Romagna Region, Northern Italy. As a result, histamine, putrescine, cadaverine, and tyramine were detected in all samples. I judge this data to be valuable because this study addresses various food safety issues involving T. mutabilis. However, I have a few comments, explained below. I hope that my comments are very useful for the improvement of this research.
Comments
(1) English language: Please ask someone who is the English native speaker to proofread your English writing. There are several unsuitable word uses in your paper.
(2) Abstract: No specific objectives are given in Abstract. The purpose should be indicated.
(3 L15. “load”: I don't know what the A stands for. Are you using “load” to mean "include or contain"? If so, I don't think “load” is common.
(4) Introduction: The Introduction is not well organized and is fragmented. Your Introduction is also too long. Please summarize better. Also, please add the following points.
(a). An outbreak of food poisoning caused by T. mutabilis.
(b). Status of T. mutabilis intake among Italians.
(5) The values of Table: The value is only shown as an average (or one measurement result?). How many replicates have been repeated to measure the number of Vibrio, BAs, BAI, and IPB contents? If possible, standard error of means should be shown.
(6) Table 1 and 2: The date of collection or purchase of each T. mutabilis should be stated.
(7) Discussion: I would like you to discuss whether there is a risk of food poisoning by Vibrio, BAs, BAI, and IPB contents, based on the amount of T. mutabilis consumed by Italians.
Author Response
In this manuscript entitled "Hazard identification related to the presence of Vibrio spp., biogenic amines, and indole producing bacteria in a non-filter feeding marine gastropod (Tritia mutabilis) commercialized on the Italian market", the authors examined possible foodborne hazards related to the accumulation of bacteria and chemical products of the decomposition in T. mutabilis from Emilia-Romagna Region, Northern Italy. As a result, histamine, putrescine, cadaverine, and tyramine were detected in all samples. I judge this data to be valuable because this study addresses various food safety issues involving T. mutabilis. However, I have a few comments, explained below. I hope that my comments are very useful for the improvement of this research.
Answer: The Authors thank the Reviewer for the positive comments on the paper and for the suggestions provided.
Comments
(1) English language: Please ask someone who is the English native speaker to proofread your English writing. There are several unsuitable word uses in your paper.
Answer: The Authors agree with the Reviewer’s comments. A native English speaker has proofread our English writing.
(2) Abstract: No specific objectives are given in Abstract. The purpose should be indicated.
Answer: The Authors agree with the Reviewer’s comments. The Abstract has been revised and specific objectives included (L 15-16).
(3) L15. “load”: I don't know what the A stands for. Are you using “load” to mean "include or contain"? If so, I don't think “load” is common.
Answer: The term “load” in this context means “abundance, quantity”, and it is largely utilized by microbiologists. Some definitions are given below:
- Bacterial load definition: Measurable quantity of bacteria in an object, organism, or organism compartment (Medical Dictionary Online, https://www.online-medical-dictionary.org/definitions-b/bacterial-load.html).
- Microbial load definition: The number and type of microorganisms contaminating an object or organism (USDA-NAL Agricultural Thesaurus and Glossary, https://agclass.nal.usda.gov/mtwdk.exe?s=1&n=1&y=0&l=60&k=glossary&t=2&w=microbial+load
(4) Introduction: The Introduction is not well organized and is fragmented. Your Introduction is also too long. Please summarize better. Also, please add the following points.
Answer: The Authors agree with the Reviewer’s comments. The introduction was revised accordingly.
(a). An outbreak of food poisoning caused by T. mutabilis.
Answer: To our best knowledge no outbreaks of food poisoning caused by T. mutabilis have ever been reported. This information was added in the Introduction (L. 78-79). The species has not been investigated from a sanitary point of view by other researchers. The only one paper was published by our group. (Serratore P., Zavatta E., Bignami G., Lorito L. 2019. Preliminary investigation on the microbiological quality of edible marine gastropods of the Adriatic Sea, Italy. Italian Journal of Food Safety, 8-7691:96-101).
(b). Status of T. mutabilis intake among Italians.
Answer: A common serving size is about 250 g, yielding approximately 50-70 g of flesh (20-30%) of T. mutabilis. Considering the mean value of BAI registered in the batches from primary production (50.45 mg kg-1) and the mean value registered for batches at retail (65.83 mg kg-1), a common serving may have a BAI content of 35 mg and 46 mg respectively. Both the values are higher than those settled for the rejection of some species of fish.
The hazard per serving was not calculated for the following reasons:
- pathogenic vibrios (some hazardous limits are indicated only by FDA) were not detected;
- the maximum amount of single BAs, except histamine, and IPB are not provided by Legislation;
- currently no precise value is indicated for BAI toxicity, but the maximum amount of BAI is indicated for some species (rejection limit) and these values were comparatively considered in the paper.
(5) The values of Table: The value is only shown as an average (or one measurement result?). How many replicates have been repeated to measure the number of Vibrio, BAs, BAI, and IPB contents? If possible, standard error of means should be shown.
Answer: In the Tables, the mean value was calculated as the result of the Vibrio, BAs, BAI, and IPB contents of each batch divided by the number of examined batches. For clarity, the standard deviation value has been now reported on a separate line. As required, the standard error was calculated.
(6) Table 1 and 2: The date of collection or purchase of each T. mutabilis should be stated.
Answer: The Authors agree with the Reviewer’s comments. The dates of collection or purchase have been added in Tables 1 and 2, and also in Table 3.
(7) Discussion: I would like you to discuss whether there is a risk of food poisoning by Vibrio, BAs, BAI, and IPB contents, based on the amount of T. mutabilis consumed by Italians.
Answer: We should be aware that legislation has its limitations (pathogenic vibrios, BAI and Indole content not considered by EU Legislation). With respect to BAs, it is designed for one single biogenic amine (histamine) but it is not the only biogenic amine responsible of toxicity, and other BAs can potentiate histamine effects. See also reply to comment 4b).
Reviewer 3 Report
Thank you for the opportunity to review this article.
The auuthors evaluated the Hazard identification related to the presence of Vibrio spp., biogenic amines, and indole producing bacteria in a non-filter
feeding marine gastropod (Tritia mutabilis) commercialized on
the Italian market.
Short remarks follow below.
The first line of Table 2 should be written in horizontal format.
Conclusion: The conclusion is weak. The authors must tell the reader what their recommendations.
Author Response
Thank you for the opportunity to review this article.
The authors evaluated the Hazard identification related to the presence of Vibrio spp., biogenic amines, and indole producing bacteria in a non-filter feeding marine gastropod (Tritia mutabilis) commercialized on the Italian market.
Short remarks follow below.
The first line of Table 2 should be written in horizontal format.
Answer: The Authors agree with the Reviewer’s suggestion. The first line of Table 2 has been written in horizontal format.
Conclusion: The conclusion is weak. The authors must tell the reader what their recommendations.
Answer: The Authors agree with the Reviewer’s comments. The Conclusions has been revised accordingly.